# Perceptions of influenza and SARS-CoV-2 vaccination among health care personnel in Thailand, 2024

Kriengkrai Prasert[1,2], Prabda Praphasiri[2,3]*, Darunee Ditsungnoen[3], Sribud Srichaijaroonpong[2], Ratchadaporn Ungcharoen[2], Anusak Kerdsin[2], Sutthichai Nakphook[1], William W. Davis[3,4], Chakrarat Pittayawonganon[5], Eva S. Bazant[6], Ann C. Moen[6], Jaymin C. Patel[4], Julie G. Carlton[4], Martha P. Montgomery[3,4]

**1** Nakhon Phanom Provincial Hospital, Nakhon Phanom, Thailand, **2** Faculty of Public Health, Chalermphrakiat Sakon Nakhon Province Campus, Kasetsart University, Sakon Nakhon, Thailand, **3** Thailand MOPH-US CDC Collaboration, Nonthaburi, Thailand, **4** U.S. Centers for Disease Control and Prevention, Atlanta, Georgia, United States of America, **5** Health Technical Office, Office of the Permanent Secretary, Ministry of Public Health, Nonthaburi, Thailand, **6** The Task Force for Global Health, Decatur, Georgia, United States of America

* hpu3@cdc.gov

## Abstract

### Background

Influenza and COVID-19 vaccinations are recommended for health care personnel (HCP) in Thailand, but uptake depends on HCP perceptions and motivations.

### Methods

To assess factors associated with self-reported influenza vaccination in the most recent season and intention to receive future COVID-19 vaccination annually, HCP from 16 hospitals across eight Thailand provinces were surveyed during December 2023 through January 2024. Additional survey variables included demographic and occupational characteristics, prior experiences with vaccination, perceptions of disease severity and vaccine safety and effectiveness. Multilevel mixed effect multivariable logistic regression was used, accounting for variability at provincial and hospital levels.

### Results

Overall, 2,180 HCP were surveyed. Three-quarters (74.8%) reported influenza vaccination in the most recent season, and 58.1% intended to receive COVID-19 vaccination annually in the future. Previous influenza vaccination was strongly associated with reported vaccination in the current season (adjusted odds ratio [aOR] 2.94, 95% confidence interval [CI] 2.68–3.23). For future COVID-19 vaccination, perceived vaccine safety was strongly associated (aOR 3.49, 95% CI 3.18–3.84). Perceived

**Data availability statement:** All relevant data are within the manuscript and its Supporting Information files.

**Funding:** The U.S. Centers for Disease Control and Prevention through the Task Force for Global Health under Cooperative Agreement 5 NU51IP0001916-02-00 funded this work. The funders had no role in study design, data collection and analysis, decision to publish, or preparation of the manuscript.

**Competing interests:** Dr. Ratchadaporn Ungcharoen reported serving as a consultant for EdPEx operations at Kasetsart University and providing invited lectures at Kasetsart University and Nakhon Phanom University. Dr. Eva S. Bazant reported receiving consulting fees through CDC cooperative agreement number: 5 NU51IP0001916-02-00 to support this work. Dr. Ann C. Moen reported serving as technical consultant for the World Health Organization and the Task Force for Global Health. All other authors report no disclosures.

disease severity was higher for COVID-19 than for influenza, but perceived vaccine safety and effectiveness were higher for influenza than for COVID-19. The most common barrier to influenza vaccination was insufficient time to get vaccinated (23.7%); whereas the most common barrier for COVID-19 vaccination was vaccine safety concern (30.0%).

## Conclusions

Improving vaccination coverage among HCP might need different approaches for influenza and COVID-19 vaccines. Improving convenience might be especially important for increasing influenza vaccination coverage, whereas providing reassurance about COVID-19 vaccine safety might be especially important for COVID-19.

## Introduction

Influenza and COVID-19 vaccinations are key measures to protect health care personnel against acute viral respiratory infections [1,2]. Additionally, health care personnel who are vaccinated are more likely to recommend vaccination to their patients [3]. Thailand Ministry of Public Health has recommended influenza vaccination to health care personnel since 2004 and recommended COVID-19 vaccination to health care personnel during the COVID-19 pandemic starting in February 2021 [4,5]. Both vaccines are offered to health care personnel for free; however, receipt of vaccination is influenced by perceptions of individual health care personnel [6]. Influenza vaccine acceptance among health care personnel in Thailand has generally been high over the last decade, but surveys have been limited to isolated geographic areas [7–9]. COVID-19 vaccination receipt was high in Thailand during the emergency phase of the pandemic, but acceptance of vaccination after the pandemic emergency phase is uncertain [10]. Understanding factors associated with influenza and COVID-19 vaccinations among health care personnel in Thailand can inform interventions to improve vaccination coverage.

Influenza and COVID-19 vaccinations are recommended for health care personnel in Thailand because of documented risk of infection, protective benefits of vaccination, and the risk of transmitting infection to patients [1,2,11,12]. COVID-19 risk for health care personnel stemmed not only from contact with patients, but more so from exposures to coworkers and contacts in the community [13]. Influenza vaccinations can reduce lost work days from sick leave during influenza season [14]. COVID-19 vaccination similarly has benefits for health care personnel, including indirect benefits of reducing SARS-CoV-2 infections among unvaccinated household members [15,16]. The procedures for offering vaccination to health care personal varies across hospitals. Some hospitals offer vaccination clinics with scheduled appointments or walk-in clinics, others bring vaccine to wards during workers' shifts, and some used combined approaches.

At the time of this survey, most health care personnel had completed the COVID-19 vaccine primary series and at least one booster dose [5,17]. Updated COVID-19

vaccines are recommended for health care workers, but vaccines are not widely available. In addition to the importance of vaccination to protect health care personnel, health care personnel are a trusted source for recommending vaccines to patients. In a survey of pregnant people in Thailand, a recommendation for influenza vaccination from a nurse or doctor was more motivating than a recommendation from the Ministry of Public Health or from family, although all three sources were important cues to action [18]. Similarly, for COVID-19 vaccination, health care providers were the most trusted source of vaccine information for Thai parents, over friends and family and government or public health information [19].

To inform interventions to improve vaccination coverage among health care personnel, health care personnel across Thailand were surveyed to assess both their willingness to be vaccinated against influenza and COVID-19 and their support for recommending vaccination to patients. The aim was to identify the characteristics, practices, and perceptions associated with influenza vaccination and intention to receive future COVID-19 vaccination.

## Methods

### Survey design

A cross-sectional survey of health care personnel was conducted in eight provinces of Thailand during December 2023 through January 2024. The survey assessed health care personnel experience with and perceptions of influenza and COVID-19 illness and vaccination. It collected information on demographic, health, and occupational characteristics of participants and receipt of influenza and COVID-19 vaccination in previous seasons. Health care personnel were asked about their perceptions of disease severity, vaccine safety, and vaccine effectiveness for influenza and COVID-19. The two primary outcome measures were self-reported receipt of influenza vaccination in the most recent season and intention to receive annual COVID-19 vaccination in the future. The survey also assessed willingness to recommend both vaccinations to patients. The survey was structured using a standard protocol and questionnaire based on the Health Belief Model to measure perceptions of susceptibility and severity of influenza disease and the benefits of, barriers to, and motivators for vaccination [20]. The questionnaire was adapted based on discussions with public health experts in Thailand and translated into Thai. Both the English and Thai versions of the questionnaire are provided in the Supporting Information (S1 and S2 Files).

### Sampling design

Hospitals were selected to represent four regions in Thailand. Two provinces were selected from each of the four regions by simple random sampling. In each province, the tertiary hospital (only one per province) was included, and one community hospital was selected by simple random sampling, for a total of 16 hospitals (Fig 1). Each hospital provided a list of all health care personnel by occupation group (physician, nurse, pharmacist, dentist, emergency medical technician/paramedic, assistant, aide, public health officer, and other personnel). Simple random sampling of health care personnel was done proportional to size of each occupational group in each hospital to reduce imbalance by occupation.

The sample size was calculated by the formula for population samples, $n = N/(1 + N(e)^2)$. Assuming a total population of health care personnel in Thailand (N) of approximately 100,000 and a desired precision (e) of 0.05 gave a preliminary sample size of 399. A design effect related to the multi-stage cluster design (personnel within a hospital within a province) was estimated by the formula $DE = 1 + m\alpha^2\mu$. Assuming a mean cluster size (m) of 30 participants, intra-class correlation (α) of 0.5 and an outcome prevalence (μ) of 0.5, a design effect of 4.75 and an adjusted sample size of 1,896 (399 x 4.75) were calculated. Lastly, accounting for 20% nonresponse, the total target sample size was 2,370 (1,896/ 0.8).

### Data collection

Health care personnel were included if they had direct patient contact (defined as face-to-face contact with a patient and might include treatment, counselling, education, or any other aspect of health care), were of Thai nationality, were aged 18

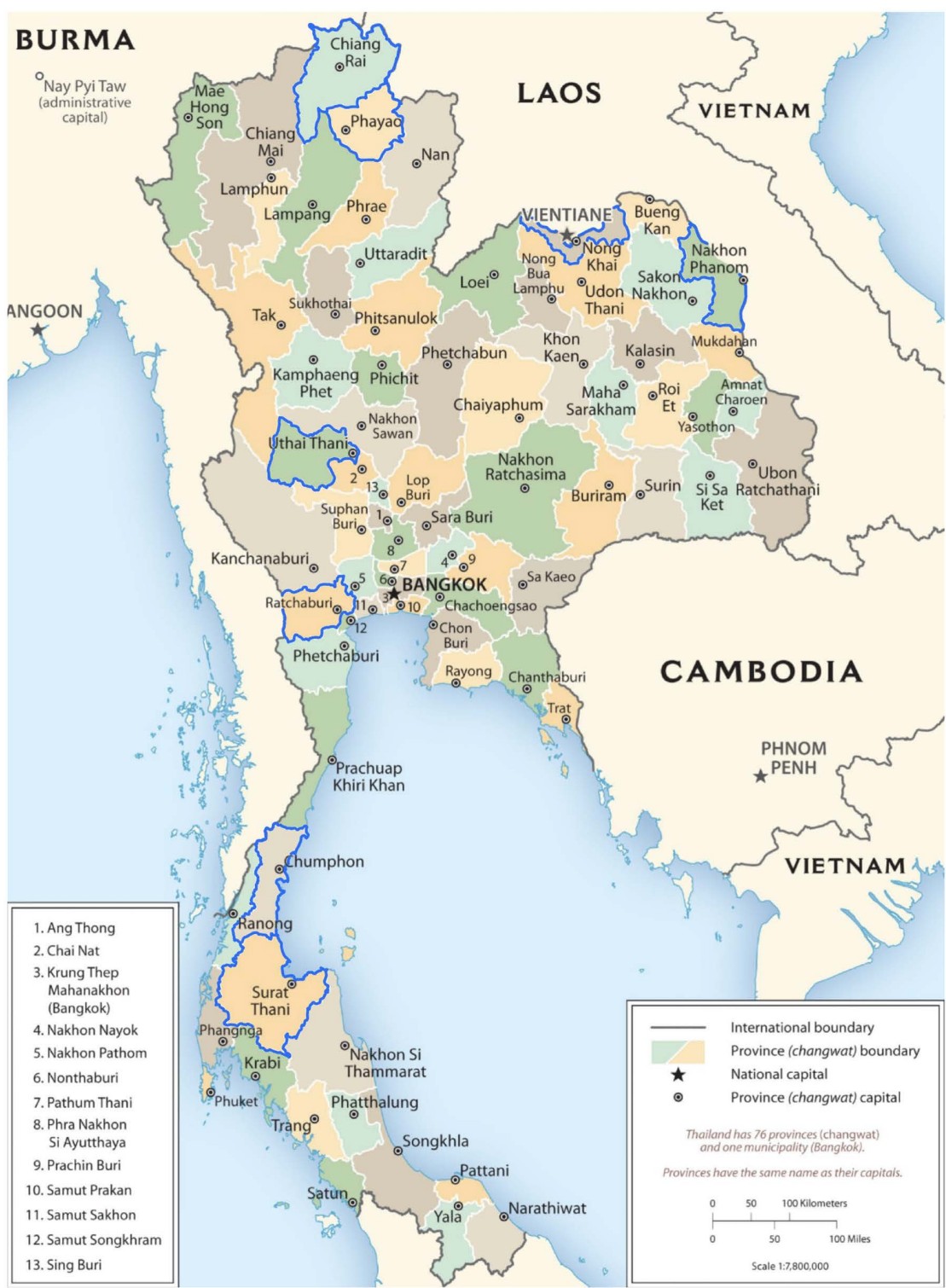

**Fig 1. Map of eight provinces (outlined in blue) with hospitals participating in the health care personnel survey on influenza and COVID-19 vaccinations—Thailand.** Two provinces in each of Thailand's four regions were selected by simple random sampling. The tertiary hospital for each province and one community hospital selected by simple random sampling were included (16 total hospitals). https://www.cia.gov/static/99eb69ecead-8f4848ee59a1c90487ed8/thailand-administrative.jpg.

years or older, and could speak and write in Thai. Health care personnel were excluded if they refused to participate in the survey or were not working at the hospital during the data collection period. Trained interviewers administered the survey questions to improve data completeness and standardize interpretation of the survey questions. Questions included multiple choice and open-ended responses; however, only multiple-choice responses were included in this analysis. Open-ended responses were used to provide additional clarification to multiple choice responses and were not systematically analyzed. Interviewers were provided with a one-day training on data collection using standard protocol language to limit variability and to ensure each question was understood before conducting interviews. Responses were recorded on paper forms and entered into an OpenDataKit database (ODK Inc, San Diego, California) [21]. Interviewers conducted double data entry and checked data for completeness and consistency prior to leaving the interview site. A data manager was responsible for data validation and data cleaning.

All survey participants provided written, informed consent. Ethical approval was received, and the study was exempted from review by the Kasetsart University Research Ethics Committee. This activity was reviewed by the U.S. Centers for Disease Control and Prevention (CDC), deemed not research, and was conducted consistent with applicable federal law and CDC policy (See, e.g., 45 CFR part 46, 21 CFR part 56; 42 USC §241(d); 5 USC §552a; 44 USC §3501 et seq.).

## Data analysis

**Variables.** The main outcomes were self-reported receipt of influenza vaccine in the most recent season, measured by an affirmative answer to the question, "Did you receive the influenza vaccine in the last season (2022)?" and intention to receive annual COVID-19 vaccination in the future by the question, "If COVID-19 vaccine becomes an annually recommended vaccine for health personnel, like the seasonal influenza vaccine, will you be vaccinated every year with COVID-19 vaccine?"

Questions on perceived disease severity, vaccine safety, and vaccine effectiveness for each population group (e.g., older adults, pregnant women) had five response options: 4 = strongly agree, 3 = agree, 2 = disagree, 1 = strongly disagree, and 0 = refuse to answer. Responses for each population group were summed into scores for each perception construct (disease severity, vaccine safety, and vaccine effectiveness). Scores were normalized to mean of 0 and standard deviation of 1. Scores for each construct were dichotomized as higher or lower than the mean value of the normalized z-score.

**Weighting.** Data were weighted to distribution of health care personnel by occupations and hospitals. A poststratification weight for each participant was developed through raking calibration procedure by occupational type and region to minimize bias in the estimates because of the disproportional representation of occupational subgroups [22]. The population totals used in raking calibration were estimated using the reported number of health care personnel in each hospital.

Weighted proportion of respondents who were vaccinated or intend to be vaccinated were calculated by enrollment setting, respondent characteristics, prior influenza and COVID-19 related practices, and key perceptions. Corresponding confidence intervals (CI) were calculated using the Korn-Graubard method [23]. Proportions in the descriptive analysis that were below pre-defined reliability criteria (e.g., based on sample size, confidence interval, or degrees of freedom) were not shown [24].

Odd ratios and 95% CI were calculated using multilevel mixed effect univariable logistic regression accounting for provincial and hospital level. Factors that were significant in the univariable analysis were included in the multivariable model. No interaction terms were included in the logistic regression model. Statistical significance was considered at level p < 0.05. All analyses were conducted in STATA, version 18 (StataCorp. 2023. Stata Statistical Software: Release 18. College Station, TX). Both the cleaned dataset and data dictionary used in this analysis are provided in the Supporting Information (S3 and S4 Files).

## Results

Overall, 2,275 health care personnel were approached, and 2,180 health care personnel participated in the survey (96% participation). Approximately three-quarters (74.8%, 95% confidence interval [CI] 72.9–76.7%) of health care personnel reported receiving influenza vaccination in the most recent season (Table 1). In unadjusted models, there was no meaningful difference in influenza vaccination coverage between provincial and district level hospitals. Differences in vaccination coverage by region, age, years in health care, and occupation were noted.

From the multivariable model, health care personnel who reported receiving influenza vaccination before the COVID-19 pandemic had 2.94 times the adjusted odds (95% CI 2.68–3.23) of influenza vaccination in the most recent season than personnel who had not received influenza vaccination before COVID-19. Among only physicians, those who reported ever diagnosing influenza or ever treating severe influenza had greater odds of influenza vaccination in the most recent season compared with physicians who did not report diagnosing or treating severe influenza (adjusted odds ratio [aOR] 5.76, 95% CI 3.56–9.31; aOR 11.02, 95%CI 6.37–19.04, respectively). Higher perceived vaccine safety (aOR 1.27, 95% CI 1.12–1.43) and perceived vaccine effectiveness (aOR 1.43, 95% CI 1.27–1.61) were both associated with increased adjusted odds of influenza vaccination. Perceived influenza severity was not associated with vaccination in unadjusted models (unadjusted OR 1.08, 95% CI 1.00–1.17), and therefore not included in multivariable modeling.

Overall, 58.1% (95% CI 56.0–60.3) of health care personnel reported that they intend to receive annual COVID-19 vaccination in the future (Table 2). Interestingly, in multivariable models, men and personnel with less than 10 years of work experience reported higher intention to receive annual COVID-19 vaccination compared with women and personnel with more years of experience, respectively (Table 2), which was the opposite direction of association from actual influenza vaccination in the most recent season (Table 1).

In the multivariable model for COVID-19 vaccination, physicians who had ever diagnosed COVID-19 but not ever treated COVID-19 had higher intention to receive annual COVID-19 vaccination (aOR 7.10, 95% CI 3.68–13.71). For all occupations, personnel with higher perceived vaccine safety had greater odds of intending to receive future COVID-19 vaccination annually (aOR 3.49, 95% CI 3.18–3.84) than personnel with lower perceived vaccine safety. Higher perceived COVID-19 disease severity (aOR 1.35, 95% CI 1.25–1.47) and perceived COVID-19 vaccine effectiveness (aOR 1.53, 95% CI 1.38–1.70) were also associated with increased adjusted odds of intent to be vaccinated. Personnel with prior experience with receiving influenza vaccination, either in the most recent season (aOR 1.46, 95% CI 1.33–1.60), prior to the COVID-19 pandemic (aOR 1.42, 95% CI 1.30–1.55), or during 2009 H1N1 pandemic (aOR 1.34, 95% CI 1.23–1.46) had increased adjusted odds of intending to receive annual COVID-19 vaccination.

The proportion of health care personnel who agreed that influenza can result in severe illness (average 82%, range 69–94% across key population groups) was lower than the proportion who perceived that COVID-19 can result in severe illness for the same key populations (average 91%, range 84–95%, p=0.002 for paired t-test comparing averages) (Fig 2). In contrast, more health care personnel agreed that influenza vaccine was safe (average 91%, range 82–96%) and effective (average 94%, range 86–97%) than agreed that COVID-19 vaccine was safe (average 75%, range 61–91%, p<0.001) and effective (average 83%, range 76–87%, p<0.001). Overall, the proportion of participants who refused to answer perception questions was low. Health care personnel perceptions of COVID-19 vaccine safety differed depending on the target population. Over 90% of health care personnel agreed that COVID-19 vaccine is safe for children ages five years and lower; whereas only 60% of health care personnel agreed that COVID-19 vaccine is safe for a pregnant person's fetus. The remainder of health care personnel were divided between disagree (20%) and did not answer (20%) regarding the safety of COVID-19 vaccine for a pregnant person's fetus.

Motivating factors for receiving or intending to receive vaccination differed between influenza and COVID-19 vaccinations (Table 3). For influenza vaccination, more than half of health care personnel reported that protecting themselves from infection (64.9%, 95%CI 62.5–67.2%), vaccination offered free at work (65.4%, 95%CI 63.0–67.8%), and receiving

**Table 1. Factors associated with receiving influenza vaccination in the last season among Thai health care personnel in January 2024.**

| | Total | Receiving influenza vaccination Weighted % (95% CI)[a] | Crude OR[b] (95%CI) | p-value | Adjusted OR[c] (95%CI) | p-value |
|---|---|---|---|---|---|---|
| Overall | 2,180 | 74.8 (72.9, 76.7) | – | – | – | – |
| Region | | | | | | |
| Central | 547 (24.8) | 68.4 (64.3, 72.3) | Ref. | – | Ref. | – |
| North | 623 (29.0) | 78.8 (75.3, 82.1) | 1.89 (0.31, 2.50) | 0.820 | 0.79 (0.27, 2.35) | 0.671 |
| South | 612 (29.2) | 72.9 (69.1, 76.5) | 0.20 (0.08, 0.48) | <0.001 | 0.15 (0.06, 0.38) | <0.001 |
| Northeast | 398 (17.0) | 80.7 (76.3, 84.5) | 1.03 (0.44, 2.41) | 0.944 | 0.73 (0.31, 1.71) | 0.463 |
| Type of hospital | | | | | | |
| District hospital | 467 (21.1) | 74.5 (70.1, 78.5) | Ref. | – | – | – |
| Provincial hospital | 1,713 (78.9) | 74.9 (72.8, 77.0) | 0.95 (0.49, 1.83) | 0.878 | – | – |
| Sex | | | | | | |
| Male | 408 (18.7) | 71.3 (66.5, 75.7) | Ref. | – | Ref. | – |
| Female | 1,772 (81.3) | 75.7 (73.5, 77.7) | 1.33 (1.20, 1.47) | <0.001 | 1.29 (1.15, 1.45) | <0.001 |
| Age group | | | | | | |
| 18–29 years | 507 (23.6) | 72.7 (68.6, 76.6) | Ref. | – | Ref. | – |
| 30–44 years | 896 (41.6) | 73.4 (70.2, 76.3) | 1.06 (0.96, 1.17) | 0.267 | 0.84 (0.73, 0.96) | 0.012 |
| 45–56 years | 760 (34.1) | 77.8 (74.7, 80.7) | 1.40 (1.25, 1.56) | <0.001 | 1.06 (0.88, 1.27) | 0.546 |
| ≥ 60 years | 17 (0.1) | —[d] | 2.38 (1.23, 4.62) | 0.010 | 1.49 (0.72, 3.07) | 0.283 |
| Years in healthcare | | | | | | |
| ≤ 10 years | 677 (31.3) | 71.8 (68.2, 75.2) | Ref. | – | Ref. | – |
| 11–20 years | 415 (19.3) | 77.4 (73.1, 81.4) | 1.45 (1.29, 1.63) | <0.001 | 1.33 (1.15, 1.53) | <0.001 |
| 21–30 years | 692 (31.7) | 74.0 (70.4, 77.4) | 1.18 (1.07, 1.31) | 0.001 | 1.02 (0.89, 1.18) | 0.761 |
| > 30 years | 396 (17.7) | 78.8 (74.5, 82.8) | 1.62 (1.43, 1.83) | <0.001 | 1.21 (0.99, 1.47) | 0.067 |
| Occupation[e] | | | | | | |
| Physician | 215 (10.1) | 78.5 (72.4, 83.8) | Ref. | – | Ref. | – |
| Nurse | 1,029 (46.2) | 77.4 (74.7, 79.9) | 0.91 (0.79, 1.05) | 0.209 | 0.76 (0.65, 0.90) | 0.001 |
| Pharmacist | 63 (3.0) | 61.1 (47.9, 73.2) | 0.41 (0.32, 0.52) | <0.001 | 0.37 (0.28, 0.48) | <0.001 |
| Dentist, emergency medical technician, paramedics | 114 (5.5) | 72.9 (63.8, 80.8) | 0.73 (0.59, 0.90) | 0.003 | 0.80 (0.63, 1.00) | 0.059 |
| Assistant/aide | 323 (15.1) | 74.2 (69.1, 78.9) | 0.71 (0.60, 0.84) | <0.001 | 0.73 (0.60, 0.87) | 0.001 |
| Public health officer | 51 (4.8) | 70.3 (55.8, 82.3) | 0.62 (0.50, 0.78) | <0.001 | 0.96 (0.75, 1.23) | 0.752 |
| Other personnel | 385 (15.3) | 70.0 (65.1, 74.6) | 0.63 (0.53, 0.74) | <0.001 | 0.79 (0.66, 0.95) | 0.012 |
| Type of patients generally treat | | | | | | |
| General | 1,058 (48.3) | 70.8 (67.9, 73.5) | Ref. | – | Ref. | – |
| Pregnant woman | 133 (6.2) | 69.6 (60.8, 77.5) | 0.87 (0.74, 1.02) | 0.081 | 0.76 (0.64, 0.91) | 0.003 |
| Children | 189 (8.6) | 79.5 (73.0, 85.0) | 1.67 (1.43, 1.95) | <0.001 | 1.33 (1.13, 1.58) | 0.001 |
| Adult with chronic disease | 388 (17.9) | 79.2 (74.6, 83.2) | 1.67 (1.48, 1.86) | <0.001 | 1.70 (1.44, 1.99) | <0.001 |
| Adult with infectious disease | 195 (8.9) | 79.9 (73.6, 85.3) | 1.53 (1.31, 1.78) | <0.001 | 1.23 (1.01, 1.49) | 0.042 |
| Older adults (above 65 years) | 217 (10.0) | 84.4 (75.4, 86.4) | 1.50 (1.29, 1.74) | <0.001 | 1.54 (1.28, 1.84) | <0.001 |
| Frontline work[f] | | | | | | |
| No | 1,277 (59.1) | 71.5 (68.9, 74.0) | Ref. | – | Ref. | – |
| Yes | 903 (40.9) | 79.6 (77.0, 82.2) | 1.47 (1.35, 1.60) | < 0.001 | 1.11 (0.97, 1.27) | 0.132 |
| Has high-risk condition(s)[g] | | | | | | |
| No | 1,680 (77.1) | 73.6 (71.4, 75.8) | Ref. | – | Ref. | – |
| Yes | 500 (22.9) | 78.9 (75.0, 82.4) | 1.48 (1.35, 1.60) | <0.001 | 1.20 (1.08, 1.34) | 0.001 |

*(Continued)*

**Table 1.** (Continued)

| | Total | Receiving influenza vaccination Weighted % (95% CI)[a] | Crude OR[b] (95%CI) | p-value | Adjusted OR[c] (95%CI) | p-value |
|---|---|---|---|---|---|---|
| **Ever diagnosed influenza** | | | | | | |
| No | 87 (40.2)[h] | 69.8 (59.0, 79.2) | Ref. | – | Ref. | – |
| Yes | 128 (59.8) | 84.4 (76.9, 90.2) | 3.31 (2.43, 4.53) | <0.001 | 5.76 (3.56, 9.31) | <0.001 |
| **Ever treated pneumonia/severe cases of influenza** | | | | | | |
| No | 69 (69.2)[h] | 66.0 (53.6, 77.0) | Ref. | – | Ref. | – |
| Yes | 146 (30.8) | 84.3 (77.3, 89.9) | 2.77 (1.44, 5.30) | 0.002 | 11.02 (6.37, 19.04) | <0.001 |
| **Ever got influenza vaccination during 2009 influenza pandemic** | | | | | | |
| No | 1,373 (63.5) | 70.1 (67.5, 72.5) | Ref. | – | Ref. | – |
| Yes | 807 (36.5) | 83.1 (80.3, 85.7) | 2.13 (1.94, 2.33) | < 0.001 | 1.63 (1.47, 1.80) | <0.001 |
| **Received influenza vaccine before onset of COVID-19** | | | | | | |
| No | 637 (29.2) | 58.7 (54.5, 62.6) | Ref. | – | Ref. | – |
| Yes | 1,543 (70.8) | 81.6 (79.5, 83.5) | 3.18 (2.92, 3.47) | < 0.001 | 2.94 (2.68, 3.23) | <0.001 |
| **Perceived influenza disease severity[i]** | | | | | | |
| Low | 1,274 (58.4) | 74.4 (71.8, 76.8) | Ref. | – | – | – |
| High | 906 (41.6) | 75.5 (72.5, 78.3) | 1.08 (1.00, 1.17) | 0.055 | – | – |
| **Perceived influenza vaccine safety[i]** | | | | | | |
| Low | 1,462 (67.1) | 72.3 (69.9, 74.6) | Ref. | – | Ref. | – |
| High | 718 (32.9) | 80.0 (76.8, 82.9) | 1.61 (1.47, 1.76) | < 0.001 | 1.27 (1.12, 1.43) | <0.001 |
| **Perceived influenza vaccine effectiveness[i]** | | | | | | |
| Low | 1,365 (64.0) | 71.5 (69.0, 73.9) | Ref. | – | Ref. | – |
| High | 785 (36.0) | 80.7 (77.7, 83.4) | 1.78 (1.63, 1.94) | < 0.001 | 1.43 (1.27, 1.61) | <0.001 |

[a]Korn-Graubard 95%CI.

[b]Multilevel mixed-effect univariable logistic regression model.

[c]Characteristics and influenza-related factors significant in univariate analysis were adjusted in the final multilevel mixed-effect multivariable logistic regression model (except 'ever diagnosis influenza' and 'ever treat pneumonia/severe cases of influenza', which were analyzed only among doctors). Statistical significance at p-value < 0.05.

[d]Estimate does not meet the National Center for Health Statistic's standards of reliability. https://www.cdc.gov/nchs/data/series/sr_02/sr02_175.pdf.

[e]Other personnel include central supply staffs, professional assistants, nutritionist, Thai traditional therapists, porter, drivers, and general service officers.

[f]Frontline work included health care personnel in general and internal medicine, intensive care units, and the emergency department. Non-frontline work included health care personnel in obstetrics & gynecology, pediatrics, radiology, surgery, and others.

[g]Including obesity, diabetes, cardiac diseases, pulmonary diseases, immunocompromised, and other high-risk conditions.

[h]Data were analyzed only among physicians.

[i]Sum score of each perception was used to construct the scores. The total score of each construct was divided into two (low and high) at the mean value of the standardized sum score.

vaccination every year (67.8%, 95%CI 65.5–70.2%) were motivating reasons. For COVID-19 vaccination, half of health care personnel similarly cited that protecting themselves from infection (51.6%, 95%CI 48.7–54.4%) was the top reason to be vaccinated but having it free at work was less frequently a motivating factor (27.8%, 95%CI 25.3–30.4). The most frequent barrier to influenza vaccination was not enough time to get vaccinated (23.7%, 95%CI 20.1–27.6%) followed by concern for side effects or adverse reactions (19.8%, 95%CI 16.5–23.5%). Concerningly, around one in seven health care personnel who were not vaccinated for influenza cited a preference for natural immunity as a reason not to be vaccinated (14.9%, 95% CI 12.0–18.2%). Few health care personnel reported concerns about influenza vaccine safety (3.1%, 95%CI 1.7–5.0%). In contrast, for COVID-19 vaccination, the highest barrier for personnel who did not intend to get a future

**Table 2. Factors associated with intent to receive future COVID-19 vaccination among Thai health care personnel in January 2024.**

| | Total | Intent to receiving annual COVID-19 vaccinations. Weighted % (95% CI)[a] | Crude OR[b] (95%CI) | p-value | Adjusted OR[c] (95%CI) | p-value |
|---|---|---|---|---|---|---|
| Overall | 2,180 | 58.1 (56.0, 60.3) | – | – | – | – |
| Hospital Region | | | | | | |
| Central | 547 (24.8) | 57.8 (53.5, 62.1) | Ref. | | Ref. | – |
| North | 623 (29.0) | 61.5 (57.5, 65.4) | 1.17 (0.65, 2.10) | 0.596 | 1.02 (0.63, 1.68) | 0.923 |
| South | 612 (29.2) | 49.3 (45.2, 53.4) | 0.36 (0.22, 0.60) | <0.001 | 0.45 (0.28, 0.71) | 0.001 |
| Northeast | 398 (17.0) | 68.1 (63.2, 72.8) | 2.37 (1.29, 4.35) | 0.005 | 1.64 (1.00, 2.67) | 0.049 |
| Type of hospital | | | | | | |
| District hospital | 467 (21.1) | 59.5 (54.8, 64.1) | Ref. | – | – | – |
| Provincial hospital | 1,713 (78.9) | 57.8 (55.4, 60.2) | 0.94 (0.61, 1.44) | 0.788 | – | – |
| Sex | | | | | | |
| Male | 408 (18.7) | 62.9 (57.9, 67.7) | Ref. | – | Ref. | – |
| Female | 1,772 (81.3) | 57.1 (54.7, 59.4) | 0.83 (0.76, 0.91) | <0.001 | 0.72 (0.65, 0.81) | <0.001 |
| Age group | | | | | | |
| 18–29 years | 507 (23.6) | 62.6 (58.2, 67.0) | Ref. | – | Ref. | – |
| 30–44 years | 896 (41.6) | 56.1 (52.7, 59.4) | 0.74 (0.68, 0.81) | <0.001 | 1.01 (0.89, 1.15) | 0.869 |
| 45–56 years | 760 (34.1) | 57.6 (54.0, 61.2) | 0.81 (0.74, 0.89) | <0.001 | 1.01 (0.86, 1.20) | 0.877 |
| ≥ 60 years | 17 (0.1) | –[d] | 0.61 (0.40, 0.93) | 0.023 | 0.61 (0.38, 0.98) | 0.040 |
| Years in healthcare | | | | | | |
| ≤ 10 years | 677 (31.3) | 66.2 (62.4, 69.9) | Ref. | – | Ref. | – |
| 11–20 years | 415 (19.3) | 51.2 (46.2, 56.2) | 0.57 (0.51, 0.63) | <0.001 | 0.61 (0.53, 0.69) | <0.001 |
| 21–30 years | 692 (31.7) | 54.2 (50.4, 58.0) | 0.62 (0.56, 0.67) | <0.001 | 0.60 (0.53, 0.69) | <0.001 |
| > 30 years | 396 (17.7) | 58.5 (53.4, 63.4) | 0.74 (0.66, 0.82) | <0.001 | 0.76 (0.63, 0.91) | 0.003 |
| Occupation[e] | | | | | | |
| Physician | 215 (10.1) | 65.8 (59.0, 72.1) | Ref. | – | Ref. | – |
| Nurse | 1,029 (46.2) | 57.3 (54.2, 60.4) | 0.70 (0.62, 0.79) | <0.001 | 0.78 (0.67, 0.90) | 0.001 |
| Pharmacist | 63 (3.0) | 54.9 (41.8, 67.5) | 0.66 (0.53, 0.83) | <0.001 | 0.90 (0.69, 1.16) | 0.405 |
| Dentist, emergency medical technician, paramedics | 114 (5.5) | 45.9 (36.5, 55.6) | 0.43 (0.36, 0.52) | <0.001 | 0.75 (0.60, 0.93) | 0.008 |
| Assistant/aide | 323 (15.1) | 65.8 (60.3, 71.0) | 1.04 (0.90, 1.20) | 0.618 | 1.52 (1.29, 1.80) | <0.001 |
| Public health officer | 51 (4.8) | 62.6 (47.9, 75.7) | 0.89 (0.73, 1.09) | 0.264 | 0.88 (0.70, 1.10) | 0.250 |
| Other personnel | 385 (15.3) | 51.6 (46.5, 56.7) | 0.58 (0.50, 0.67) | <0.001 | 0.74 (0.63, 0.87) | <0.001 |
| Type of patients generally treat | | | | | | |
| General | 1,058 (48.3) | 54.4 (51.3, 57.5) | Ref. | – | Ref. | – |
| Pregnant woman | 133 (6.2) | 56.5 (47.6, 65.0) | 1.05 (0.90, 1.21) | 0.547 | 0.99 (0.84, 1.17) | 0.900 |
| Children | 189 (8.6) | 65.3 (58.0, 72.0) | 1.52 (1.33, 1.73) | <0.001 | 1.64 (1.41, 1.91) | <0.001 |
| Adult with chronic disease | 388 (17.9) | 58.6 (53.5, 63.6) | 1.27 (1.16, 1.40) | <0.001 | 0.81 (0.70, 0.93) | 0.003 |
| Adult with infectious disease | 195 (8.9) | 66.3 (59.2, 72.9) | 1.50 (1.32, 1.71) | <0.001 | 0.96 (0.81, 1.14) | 0.672 |
| Older adults (above 65 years) | 217 (10.0) | 62.9 (55.9, 69.4) | 1.40 (1.24, 1.58) | <0.001 | 0.85 (0.73, 0.99) | 0.041 |
| Frontline work[f] | | | | | | |
| No | 1,277 (59.1) | 54.4 (51.6, 57.2) | Ref. | – | Ref. | – |
| Yes | 903 (40.9) | 63.5 (60.3, 66.7) | 1.44 (1.34, 1.55) | <0.001 | 1.62 (1.44, 1.82) | <0.001 |

*(Continued)*

**Table 2.** (Continued)

| | Total | Intent to receiving annual COVID-19 vaccinations. Weighted % (95% CI)[a] | Crude OR[b] (95%CI) | p-value | Adjusted OR[c] (95%CI) | p-value |
|---|---|---|---|---|---|---|
| Has high-risk conditions[g] | | | | | | |
| No | 1,680 (77.1) | 57.2 (54.8, 60.0) | Ref. | – | – | – |
| Yes | 500 (22.9) | 61.2 (56.8, 65.5) | 1.18 (1.08, 1.28) | <0.001 | 0.97 (0.89, 1.07) | 0.585 |
| Ever diagnosed COVID-19 | | | | | | |
| No | 21 (9.9)[h] | 43.2 (22.0, 66.5) | Ref. | – | Ref. | – |
| Yes | 194 (90.1) | 68.3 (61.2, 74.7) | 2.83 (1.16, 7.16) | 0.029 | 7.10 (3.68, 13.71) | <0.001 |
| Ever treated COVID-19 cases | | | | | | |
| No | 61 (28.2)[h] | 64.0 (50.7, 76.0) | Ref. | – | – | – |
| Yes | 154 (71.8) | 66.5 (58.4, 73.9) | 1.11 (0.65, 2.05) | 0.731 | – | – |
| Received influenza vaccination during 2009 influenza pandemic | | | | | | |
| No | 1,373 (63.5) | 54.6 (51.9, 57.3) | Ref. | – | Ref. | – |
| Yes | 807 (36.5) | 64.3 (60.8, 67.6) | 1.53 (1.42, 1.65) | < 0.001 | 1.34 (1.23, 1.46) | <0.001 |
| Ever received influenza vaccination in most recent season | | | | | | |
| No | 545 (25) | 45.9 (41.5, 50.2) | Ref. | | Ref. | |
| Yes | 1,635 (75) | 62.3 (59.9, 64.7) | 1.86 (1.71, 2.01) | < 0.001 | 1.46 (1.33, 1.60) | < 0.001 |
| Received influenza vaccine before onset of COVID-19 | | | | | | |
| No | 637 (29.2) | 52.1 (48.1, 56.1) | Ref. | – | Ref. | – |
| Yes | 1,543 (70.8) | 60.7 (58.2, 63.2) | 1.52 (1.41, 1.64) | < 0.001 | 1.42 (1.30, 1.55) | <0.001 |
| Perceived COVID-19 disease severity[i] | | | | | | |
| Low | 1,186 (54.4) | 51.7 (48.8, 54.6) | Ref. | – | Ref. | – |
| High | 994 (45.6) | 65.8 (62.7, 68.8) | 1.84 (1.71, 1.97) | < 0.001 | 1.35 (1.25, 1.47) | <0.001 |
| Perceived COVID-19 vaccine safety[i] | | | | | | |
| Low | 748 (34.3) | 34.5 (31.0, 38.0) | Ref. | – | Ref. | – |
| High | 1,432 (65.7) | 70.3 (67.8, 72.8) | 4.52 (4.19, 4.88) | < 0.001 | 3.49 (3.18, 3.84) | < 0.001 |
| Perceived COVID-19 vaccine effectiveness[i] | | | | | | |
| Low | 559 (25.6) | 36.4 (33.4, 40.6) | Ref. | – | – | – |
| High | 1,621 (74.4) | 65.6 (63.2, 67.9) | 3.18 (2.93, 3.45) | < 0.001 | 1.53 (1.38, 1.70) | < 0.001 |

[a]Korn-Graubard 95%CI.

[b]Multilevel mixed-effect univariable logistic regression model.

[c]Characteristics and influenza-related factors significant in univariate analysis were adjusted in the final multilevel mixed-effect multivariable logistic regression model (except 'ever diagnosis influenza' and 'ever treat pneumonia/severe cases of influenza', which were analyzed only among doctors). Statistical significance at p-value < 0.05.

[d]Estimate does not meet the National Center for Health Statistic's standards of reliability. https://www.cdc.gov/nchs/data/series/sr_02/sr02_175.pdf.

[e]Other personnel include central supply staffs, professional assistants, nutritionist, Thai traditional therapists, porter, drivers, and general service officers.

[f]Frontline work included health care personnel in general and internal medicine, intensive care units, and the emergency department. Non-frontline work included health care personnel in obstetrics & gynecology, pediatrics, radiology, surgery, and others.

[g]Including obesity, diabetes, cardiac diseases, pulmonary diseases, immunocompromised, and other high-risk conditions.

[h]Data were analyzed only among physicians.

[i]Sum score of each perception was used to construct the scores. The total score of each construct was divided into two (low and high) at the mean value of the standardized sum score.

COVID-19 vaccination was concerns about vaccine safety (29.7%, 95%CI 25.8–33.7%) followed by vaccine effectiveness (25.3% 95%CI 21.7–29.3%) and a preference for natural immunity (17.7%, 95% CI 14.6–21.2%).

Health care personnel were asked about whether they would recommend vaccination to patients. For influenza, 96.2% (95%CI 95.3–97.0%) of health care personnel would recommend influenza vaccination to patients who have an indication

**Do you agree with the statement, "Influenza can lead to hospitalizations, ICU admissions and/or death" for each of the following groups?**

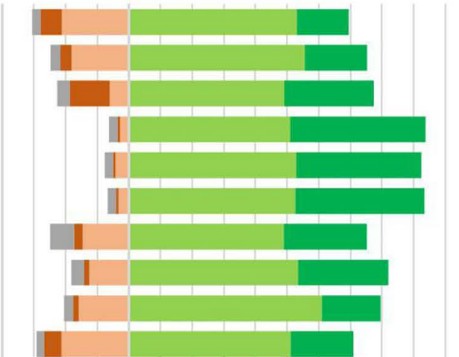

Me/Self
School-aged children (5-16 years old)
Children 5 years and lower
Immunocompromised
Individuals aged 65 years and over
Person with underlying chronic condition
A pregnant woman's fetus
Pregnant woman
People in general population
Health workers

**Do you agree with the statement, "COVID-19 can lead to hospitalizations, ICU admissions and/or death" for each of the following groups?**

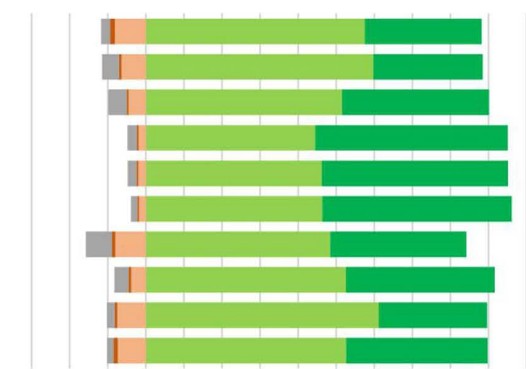

**Do you agree that influenza vaccine is safe for each of the following groups?**

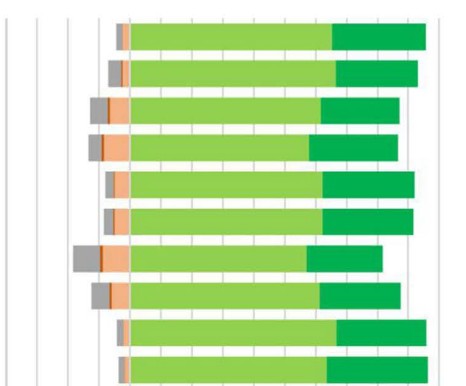

Me/Self
School-aged children (5-16 years old)
Children 5 years and lower
Immunocompromised
Individuals aged 65 years and over
Person with underlying chronic condition
A pregnant woman's fetus
Pregnant woman
People in general population
Health workers

**Do you agree that COVID-19 vaccine is safe for each of the following groups?**

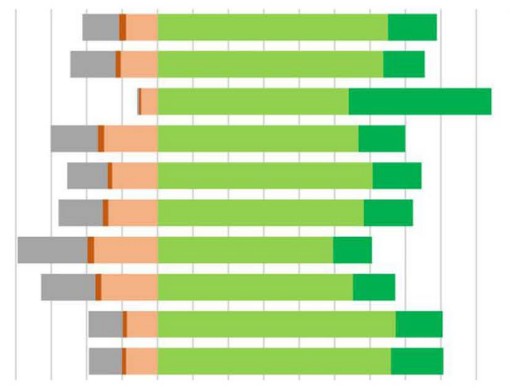

**Do you agree with the statement, "Getting the influenza vaccine can reduce the chances of becoming severely ill (i.e., requiring hospitalization) with influenza" for each of the following groups?**

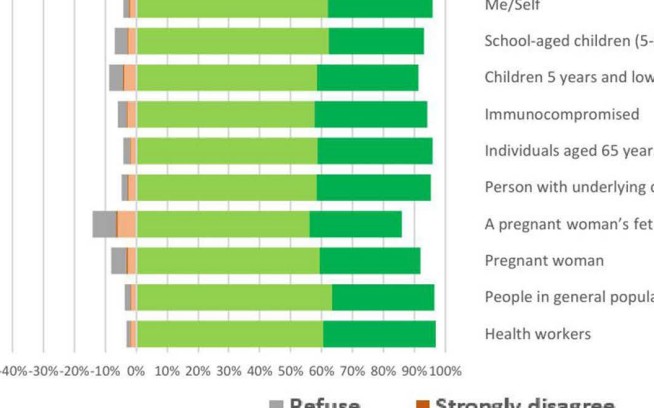

Me/Self
School-aged children (5-16 years old)
Children 5 years and lower
Immunocompromised
Individuals aged 65 years and over
Person with underlying chronic condition
A pregnant woman's fetus
Pregnant woman
People in general population
Health workers

**Do you agree with the statement, "Getting the COVID-19 vaccine can reduce the chances of becoming severely ill (i.e., requiring hospitalization) with COVID-19" for each of the following groups?**

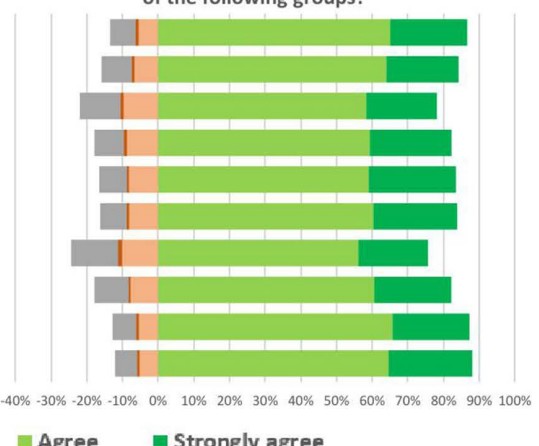

■ Refuse   ■ Strongly disagree   ■ Disagree   ■ Agree   ■ Strongly agree

**Fig 2. Perceived disease severity, vaccine safety, and vaccine effectiveness of Thai health care workers on influenza and COVID-19 in January 2024.**

**Table 3. Reasons for nonreceipt and receipt of influenza & COVID-19 vaccination among Thai healthcare workers in January 2024.**

| Reasons to receive vaccination[b] | Vaccinated against influenza (n = 1,635) | | Intend to receive annual COVID-19 vaccination (n = 1,268) | |
|---|---|---|---|---|
| | n | Weighted % (95%CI)[a] | n | Weighted % (95%CI)[a] |
| To protect myself from infection | 1,058 | 64.9 (62.5, 67.2) | 653 | 51.6 (48.7, 54.4) |
| It was offered for free at my job (not mandatory) | 1,066 | 65.4 (63.0, 67.8) | 354 | 27.8 (25.3, 30.4) |
| It was mandatory at my job | 310 | 18.8 (16.9, 20.7) | 79 | 6.3 (5.0, 7.9) |
| To protect my patients | 600 | 36.5 (34.1, 38.9) | 63 | 5.0 (3.8, 6.3) |
| To protect my family | 713 | 43.1 (40.7, 45.6) | 58 | 4.6 (3.5, 5.9) |
| It was easy/offered at my workplace | 781 | 47.7 (45.2, 50.2) | 27 | 2.1 (1.4, 3.1) |
| It was recommended by my Ministry of health or my organization | 615 | 38.1 (35.7, 40.6) | 21 | 1.6 (1.0, 2.5) |
| It was free (outside my workplace) | 689 | 41.8 (39.4, 44.3) | 1 | 0.01 (0.00, 0.04) |
| I receive the vaccine every year | 1,108 | 67.8 (65.5, 70.2) | | Not asked |
| Others[c] | 18 | 1.1 (0.6, 1.7) | 12 | 1.0 (0.5, 1.7) |
| Reasons to not receive vaccination[b] | Not vaccinated against influenza (n = 545) | | Do not intend to receive annual COVID-19 vaccination (n = 559) | |
| | n | Weighted % (95%CI)[a] | n | Weighted % (95%CI)[a] |
| Do not believe the vaccine is safe | 16 | 3.1 (1.7, 5.0) | 167 | 29.7 (25.8, 33.7) |
| Do not believe to protect me from infection | 32 | 6.0 (4.1, 8.4) | 139 | 25.3 (21.7, 29.3) |
| Prefer natural immunity | 84 | 14.9 (12.0, 18.2) | 103 | 17.7 (14.6, 21.2) |
| Do not believe in vaccination | 19 | 3.7 (2.2, 5.7) | 39 | 7.1 (5.1, 9.6) |
| Did not have time to go get vaccination | 129 | 23.7 (20.1, 27.6) | 13 | 2.2 (1.1, 3.8) |
| Do not believe be susceptible to COVID-19/influenza illness | 39 | 7.3 (5.2, 9.9) | 10 | 1.8 (0.9, 3.3) |
| It is not recommended by MOPH or my organization | 5 | 0.9 (0.3, 2.2) | 5 | 0.9 (0.3, 2.0) |
| The vaccine is not available for HCWs in my country | 3 | 0.5 (0.1, 1.5) | 4 | 0.9 (0.2, 2.5) |
| Did not want to pay or vaccination (not free at work) | 23 | 4.0 (2.5, 6.0) | 3 | 0.7 (0.1, 2.0) |
| Vaccines have side effects or adverse reactions | 106 | 19.8 (16.5, 23.5) | | Not asked |
| Vaccines make me sick | 87 | 17.1 (13.9, 20.7) | | Not asked |
| Don't like vaccination, afraid of getting hurt | 52 | 9.6 (7.2, 12.5) | | Not asked |
| A small number of vaccines, not enough | 28 | 4.9 (3.2, 7.1) | | Not asked |
| Vaccination is inconvenient or difficulties in receiving service | 12 | 2.3 (1.1, 4.1) | | Not asked |
| Others[d] | 78 | 14.1 (11.3, 17.4) | 76 | 13.8 (11.0, 17.0) |

Abbreviations: CI confidence interval.

[a]Korn-Graubard 95%CI.

[b]More than 1 reason could be selected.

[c]Examples for influenza – to reduce severity of disease, had bad experience from previous infection. Examples for COVID-19 – confidence in vaccine & reduce risk of infection.

[d]Examples for influenza – ever had side effects, I am not available, I am got COVID-19 infection, protect from no pharmaceutical methods. Examples for COVID-19 – afraid of side effects and complications; COVID-19 has changed to mild severity, no report of long-term effects.

for vaccination. For COVID-19, 68.2% (95%CI 66.2–70.2%) of health care personnel would recommend annual vaccination to patients with an indication for vaccination. Lastly, health care personnel were asked, "If influenza and COVID-19 vaccination were offered together, would you accept both?" Only 42.3% (95% CI 40.2–44.5%) would receive both vaccinations together. Around one in four (24.6%, 95% CI 22.8–26.5%) would accept influenza vaccination only. Fewer than 10% reported that they would not receive either (8.0%, 95% CI 6.8–9.3%) or would accept COVID-19 vaccination only (1.9%, 95% CI, 1.3–2.5%). Almost one-quarter (23.2%, 95% CI 21.4–25.1%) reported they were unsure.

## Discussion

Overall, willingness to receive and recommend influenza and COVID-19 vaccines was high among health care personnel in Thailand, but there were some differences in attitudes between the two vaccines. Willingness for annual COVID-19 vaccinations in the future was lower than receipt of influenza vaccination in the most recent season, despite higher perceived disease severity for COVID-19 than for influenza. Although more than half of health care personnel (58%) intended to receive annual COVID-19 vaccination in the future, intention correlates with, but does not perfectly predict, future vaccination behavior [6,25]. Most health care personnel reported that they were willing to recommend vaccination to patients, but again support was higher for influenza than for COVID-19 vaccination. These findings could be explained by the observed findings of increased concerns about COVID-19 vaccine safety, lower perceived COVID-19 vaccine effectiveness relative to influenza vaccines, or increased familiarity with influenza vaccination. Increased willingness to receive influenza vaccination compared with COVID-19 vaccination could also be related to vaccine availability, as COVID-19 vaccines are not as widely available as influenza vaccines, which could be explored further in future studies.

Other published studies of influenza vaccination coverage among health care personnel in Asia have had wide variability. A review of 35 published estimates of influenza vaccination coverage among health care personnel from 9 Asian countries reported a median coverage of 38% from studies published from 2008–2018 [26]. A more recent review estimated 28% influenza vaccination coverage among health care personnel in studies published from Asia [27]; however, the authors noted high heterogeneity in estimates across studies. Most papers in the two reviews were published with data from countries in East Asia; fewer studies have published data from Southeast Asia, mainly Singapore and Malaysia. A study from a single hospital center in Singapore reported higher influenza vaccination coverage of 82% [28]. Similar to the current study, the authors in Singapore reported that perceived severity of influenza and perceived vaccine safety were associated with vaccine acceptance. A qualitative study in Singapore reported that concerns of vaccine safety and perceived efficacy were common themes in focus group discussions with 73 health care personnel from three hospitals [29]. In Malaysia, estimated influenza vaccination coverage from 3 hospitals in 2015 had lower coverage, 51% [30]. Although fewer studies have been published on COVID-19 vaccination coverage compared with influenza, one survey of health care personnel in six Asian countries found high willingness (95%) to receive COVID-19 vaccine [31]; however, the survey was conducted in December 2020 during the pandemic emergency phase, just prior to the availability of vaccine. More studies are needed to assess the dynamic nature of health care personnel perspectives on COVID-19 vaccination in Southeast Asia in the post-emergency phase.

Based on the findings from this survey, improving respiratory vaccination coverage among health care personnel in Thailand might need separate approaches for influenza and COVID-19. To improve influenza vaccination coverage, vaccination programs might consider improving access or convenience to address the largest barrier, which was insufficient free time to get vaccinated. Strategies to improve convenience of influenza vaccination for health care personnel have been shown to improve vaccination coverage [25,32]. For example, one hospital in Germany used mobile vaccination teams and offered vaccination at convenient times to increase vaccination coverage among health care personnel [33]. One systematic review found that interventions with multiple approaches were more effective than single-approach interventions [34]. Efforts to improve vaccination convenience could be combined with educational campaigns to address concerns about vaccine side effects, which was the second most common reported barrier to influenza vaccination.

Efforts to improve COVID-19 vaccination uptake among health care personnel in Thailand should consider addressing vaccine safety and effectiveness concerns. Low perceived COVID-19 vaccine safety was the factor most strongly associated with intent to receive future COVID-19 vaccination, and concern for vaccine safety was the highest reported reason for not receiving COVID-19 vaccination. A survey of parents in Thailand during October and November 2021, similarly found that side effects and vaccine safety were the top two concerns for parents about vaccinating their children [19]. Over time, ongoing surveillance systems to monitor vaccine safety have provided robust evidence that COVID-19 vaccines are safe [35]. One survey among health care personnel in Thailand found that acceptance of COVID-19 vaccine improved from the period before vaccine was available to the period after vaccine became available [36]. This is

consistent with findings in the current survey that prior experience with receiving influenza vaccination was associated with higher acceptance of influenza and COVID-19 vaccinations. An earlier survey of nurses in Thailand similarly found that nurses who had previously been vaccinated against influenza were more likely to accept COVID-19 vaccination [37]. Taken together, these findings suggest that vaccine acceptance might improve with increasing familiarity and experience.

The COVID-19 vaccine safety topic for which there seemed to be the greatest uncertainty among Thai health care personnel was for fetal outcomes. Surveys and interviews in Thailand showed that there is similar uncertainty among pregnant people about COVID-19 vaccination safety for themselves and their fetuses, such as concerns for miscarriage or premature birth [38]. Numerous studies have shown that COVID-19 vaccination is not associated with poor maternal-fetal outcomes [39–44]. There is a need to translate and share this safety information with health care personnel so that they can provide strong vaccination recommendations to pregnant people.

For vaccine effectiveness, several studies have demonstrated the effectiveness of COVID-19 vaccines in Thailand against infection, severe disease, and hospitalization [45–47]. However, vaccine effectiveness is not static, and ongoing studies are needed to understand how vaccine effectiveness varies with changes in population immunity, SARS-CoV-2 viral evolution, and different vaccine formulations. These studies will inform COVID-19 vaccine recommendations in Thailand and can be communicated to health care personnel to help them make informed decisions about vaccination for themselves and their patients.

Several combined influenza and COVID-19 vaccine candidates are undergoing phase 3 clinical trials [48–50]. A combined SARS-CoV-2 and influenza vaccine might improve convenience for health care personnel. Alternatively, health care personnel who are unwilling to receive COVID-19 vaccine might be more likely to decline vaccination if the combined vaccine is the only vaccine offered, thereby lowering influenza vaccination coverage. Continuing to offer separate influenza and COVID-19 vaccines at the same time as a combined vaccine would offer the most flexibility to meet different patient needs but adds complication to the logistics of vaccine ordering, storage, distribution, and administration. Additional work is needed to understand acceptance of a combined influenza and COVID-19 vaccine among health care personnel, to keep health care personnel updated on the effectiveness and safety of combined vaccines, and to address potential concerns.

The survey is subject to several limitations. Although a diverse geographic range was included, provinces and hospitals were selected by convenience sampling, so the results might not be generalizable to all hospitals in Thailand. Survey weights allowed for a more valid representation of health care personnel across all occupational groups. For perception questions, participants who refused to answer were assigned a score of 0, as we assumed this response corresponded with low perceived disease severity, vaccine safety, or vaccine effectiveness; however, this response is ambiguous, and a score of 0 might not be appropriate in all cases. Overall, the proportion of participants who refused to answer was small (<10%) for most perception questions. Vaccination status was self-reported, which could be subject to misclassification or recall bias. Trained data collectors who administered the survey in a private space with anonymity were used to try to minimize the effect of social desirability bias; however, residual social desirability bias could remain and would overestimate the willingness to receive or recommend influenza and COVID-19 vaccinations.

## Conclusion

Overall vaccination acceptance was generally high among health care personnel in Thailand, both in accepting vaccination for themselves and in recommending vaccination to patients. Vaccination acceptance among health care personnel seemed to be higher for influenza than for COVID-19. Tailored approaches might be considered for improving vaccination coverage among health care personnel. To improve influenza vaccination coverage, vaccination programs could focus on improving convenience. COVID-19 vaccination campaigns could be accompanied by educational campaigns tailored to the characteristics of health care personnel to share information about vaccination safety and effectiveness, particularly in relation to the safety of COVID-19 vaccines for fetuses.

## Supporting information

**S1 File. Questionnaire in English.** Survey instrument developed for this study.
(PDF)

**S2 File. Questionnaire in Thai.** Translated version used for interviews.
(PDF)

**S3 File. KAP dataset.** Cleaned, anonymized dataset used for statistical analysis.
(XLSX)

**S4 File. Data dictionary.** Variable names and definitions used in the dataset.
(XLSX)

## Acknowledgments

The authors would like to acknowledge Dr. Sawita Srisawat (Faculty of Public Health, Kasetsart University Chalermphra-kiat Sakon Nakhon Province Campus, Sakon Nakhon Province, Thailand), Mr. Suriya Naosri, and Miss Kanlaya Sornwong (Influenza & Vaccine Study Center, Nakhon Phanom Hospital) for helping with project coordination during data collection period.

   **Disclaimer:** The findings and conclusions in this article are those of the authors and do not necessarily represent the official position of the U.S. Centers for Disease Control and Prevention.

## Author contributions

**Conceptualization:** Prabda Praphasiri, Kriengkrai Prasert, Sutthichai Nakphook, William W. Davis, Ann C. Moen, Julie G. Carlton, Martha P. Montgomery.

**Data curation:** Prabda Praphasiri, Kriengkrai Prasert, Darunee Ditsungnoen, Sribud Srichaijaroonpong, Ratchadaporn Ungcharoen, Anusak Kerdsin, Sutthichai Nakphook, Martha P. Montgomery.

**Formal analysis:** Prabda Praphasiri, Kriengkrai Prasert.

**Funding acquisition:** Sribud Srichaijaroonpong, Anusak Kerdsin, Eva S. Bazant, Ann C. Moen.

**Investigation:** Prabda Praphasiri, Kriengkrai Prasert, Darunee Ditsungnoen, Sribud Srichaijaroonpong, Ratchadaporn Ungcharoen, Sutthichai Nakphook, William W. Davis, Chakrarat Pittayawonganon, Martha P. Montgomery.

**Methodology:** Prabda Praphasiri, Kriengkrai Prasert, Sribud Srichaijaroonpong, Ratchadaporn Ungcharoen, Sutthichai Nakphook, William W. Davis, Eva S. Bazant, Ann C. Moen, Jaymin C. Patel, Julie G. Carlton, Martha P. Montgomery.

**Project administration:** Prabda Praphasiri, Kriengkrai Prasert, Darunee Ditsungnoen, Anusak Kerdsin, Ann C. Moen, Martha P. Montgomery.

**Resources:** Julie G. Carlton, Martha P. Montgomery.

**Supervision:** Anusak Kerdsin, William W. Davis, Chakrarat Pittayawonganon, Ann C. Moen, Martha P. Montgomery.

**Validation:** Prabda Praphasiri, Kriengkrai Prasert, Darunee Ditsungnoen, Ratchadaporn Ungcharoen, Anusak Kerdsin, Eva S. Bazant, Ann C. Moen, Jaymin C. Patel, Julie G. Carlton, Martha P. Montgomery.

**Writing – original draft:** Kriengkrai Prasert, Martha P. Montgomery.

**Writing – review & editing:** Prabda Praphasiri, Kriengkrai Prasert, Darunee Ditsungnoen, Sribud Srichaijaroonpong, Ratchadaporn Ungcharoen, Anusak Kerdsin, Sutthichai Nakphook, William W. Davis, Chakrarat Pittayawonganon, Eva S. Bazant, Ann C. Moen, Jaymin C. Patel, Julie G. Carlton, Martha P. Montgomery.

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
