## [Decision Letter · Decision Letter 0]

2 Apr 2025

PONE-D-24-58516Perceptions of influenza and SARS-CoV-2 vaccination among health care personnel in Thailand, 2024PLOS ONE

Dear Dr. Praphasiri,

Thank you for submitting your manuscript to PLOS ONE. After careful consideration, we feel that it has merit but does not fully meet PLOS ONE’s publication criteria as it currently stands. Therefore, we invite you to submit a revised version of the manuscript that addresses the points raised during the review process.

We look forward to receiving your revised manuscript.

Kind regards,

Sirwan Khalid Ahmed

Academic Editor

PLOS ONE

 [The U.S. Centers for Disease Control and Prevention through the Task Force for Global Health under Cooperative Agreement 5 NU51IP0001916-02-00 funded this work.]. 

Reviewers' comments:

Reviewer's Responses to Questions

**Comments to the Author**

1. Is the manuscript technically sound, and do the data support the conclusions?

Reviewer #1: Yes

Reviewer #2: Yes

2. Has the statistical analysis been performed appropriately and rigorously? 

Reviewer #1: Yes

Reviewer #2: Yes

3. Have the authors made all data underlying the findings in their manuscript fully available?

Reviewer #1: Yes

Reviewer #2: Yes

4. Is the manuscript presented in an intelligible fashion and written in standard English?

Reviewer #1: Yes

Reviewer #2: Yes

5. Review Comments to the Author

Reviewer #1: Abstract

The abstract is good, while there are few points to be better and understandable abstract.

Firstly, Please avoid from using the pronouns such as we.

Secondly, divide the sections such as background and aim or objective, then method, result, and discussion.

I mean mention all the parts separately.

Keywords: they are good no need to any change.

Introduction:

The first sentence is very complex please make it simpler to understand.

Sentences 57 to 58 instead of uptake write receiving, since uptake is not academic.

Sentence 73, In Thailand, annual influenza vaccination has been recommended for health care personnel since 2004, I think you mention it twice, look the previous sentences.

Method

Please do it separate as study design, study setting, and so on. It is better to readers.

It is better to present a diagram about selecting the provinces and hospitals.

While logistic regression models are the correct ones to use, the research does not in fact test for interactions (e.g., how gender and years of experience interact in predicting vaccine intent).

Missing response rates: It is not known how many of the healthcare workers declined to participate or whether non-responders differed substantially from responders.

Result

It is good presented.

Discussion

It is well organised while just avoid from using we. And the discussion appropriately identifies incentives and barriers but could make more comparisons with foreign literature (e.g., comparing Thai rates to rates in other countries in Asia or the world).

While the study does acknowledge certain limitations, it does not take into account social desirability bias, wherein healthcare professionals may have overreported their intention to get vaccinated in an attempt to fit in with public health needs. It also does not take into account the impact of vaccine availability on attitudes. If COVID-19 vaccines were more accessible, it is unclear if intent to vaccinate would increase.

Conclusion

It is good

References

It is good and they are in the same style no need to change.

Reviewer #2: General comment

This study assessed factors associated with influenza vaccination in the most recent season and intention to receive annual COVID-19 vaccination among health care personnel (HCP) in Thailand, using data from interviews conducted across 16 hospitals in eight provinces.

Given that both vaccinations are recommended for HCP in Thailand, the findings are relevant and timely, particularly in the context of efforts to improve vaccine uptake.

The manuscript is clearly written, appropriately structured, and employs suitable methodology.

While the manuscript is generally well-prepared, I would like to offer a few comments and questions that may help further improve the clarity and impact of the work.

Major comments

1. I understand that the survey was conducted through interviews, but some aspects of the survey methodology remain unclear. For example, it is not clear whether the responses were selected from predefined choices or provided as open-ended answers. If predefined options were used, it seems that data could have been collected through self-administered questionnaires or an online survey instead.

It would be helpful if the authors could clarify why interviews were chosen as the method of data collection.

2. The manuscript provides the rationale for the sample size calculation and the actual number of participants included; however, it would be important to also report how many individuals were initially invited to participate and how many agreed to be interviewed (i.e., the response rate).

3. The authors state that the data were weighted according to the distribution of health care personnel by occupation and hospital. However, it is not clear whether other factors—such as age, sex, or the type of patients generally treated—were also adjusted for.

Would it be necessary to consider adjusting for these variables as well?

Additionally, rather than adjusting for occupation, it may be worth considering subgroup analyses by occupational category to explore whether associations differ across groups, which could inform tailored strategies for different types of HCP. What are the authors’ thoughts on this approach?

4. The authors state that responses were summed into scores for each perception construct using the response options ranging from 0 to 4, where 0 indicates "refuse to answer." If the numerical values were used as-is, the inclusion of 0 for "refuse to answer" may affect the validity of the score calculation.

Could the authors clarify how the scores were constructed, and specifically how "refuse to answer" were handled in the scoring process?

5. Table 3 includes “It was offered for free at my job (not mandatory)” and “It was free,” but the difference between them was not clear to me. It may be helpful to provide a clearer explanation in the manuscript to avoid potential confusion for readers.

Minor comments

1. The term "vaccine effectiveness" is used throughout the manuscript, but "vaccine efficacy" appears on page 8, line 152. Are the authors intentionally distinguishing between the two, or should the terminology be made consistent?

2. I did not fully understand what you meant by “Proportions in the descriptive analysis that were below the National Center for Health Statistics reliability criteria were not shown [24]." on page 8, lines 164-165. Could you kindly explain?

3. On page 18, line 300, it appears that the word "vaccination" may be missing after "COVID-19." Could the authors confirm and revise if necessary?

6. PLOS authors have the option to publish the peer review history of their article (what does this mean? ). If published, this will include your full peer review and any attached files.

**Do you want your identity to be public for this peer review?** For information about this choice, including consent withdrawal, please see our Privacy Policy .

Reviewer #1: **Yes: ** Assisst. Prof. Dr. Kochr Ali Mahmoo

Reviewer #2: No

---

## [Author Response · Author response to Decision Letter 1]

29 May 2025

Reviewer #1: Abstract

The abstract is good, while there are few points to be better and understandable abstract.

Firstly, Please avoid from using the pronouns such as we.

We revised the abstract to remove “we” by changing from active to passive voice (lines 28-29 & 32). We made the same changes in the body of the manuscript (lines 84-88, 116-121, 312, 319, 367, 397-404).

Secondly, divide the sections such as background and aim or objective, then method, result, and discussion.

I mean mention all the parts separately.

We divided and labeled the sections of the abstract as background, methods, results, and conclusions.

Introduction:

The first sentence is very complex please make it simpler to understand.

We simplified the first sentence for clarity by separating it into two sentences and simplifying the language (line 52). It now reads, “Influenza and COVID-19 vaccinations are key measures to protect health care personnel against acute viral respiratory infections [1, 2]. Additionally, health care personnel who are vaccinated are more likely to recommend vaccination to their patients [3].”

Sentences 57 to 58 instead of uptake write receiving, since uptake is not academic.

We replaced “uptake” with “receipt of vaccination” on lines 56-57 and 59-60.

Sentence 73, In Thailand, annual influenza vaccination has been recommended for health care personnel since 2004, I think you mention it twice, look the previous sentences.

The reviewer is correct. We mention in multiple paragraphs that influenza vaccination is recommended to health care personnel. The mention on line 73 adds that the recommendation was first made in 2004. To streamline the information and reduce redundancy, we moved this information above to line 54, which now reads, “Thailand Ministry of Public Health has recommended influenza vaccination to health care personnel since 2004…”

Method

Please do it separate as study design, study setting, and so on. It is better to readers.

We separated the methods and organized the paragraphs into ‘Survey design’, ‘Sampling design’, and ‘Data collection’. These precede the ‘Data analysis’ section, which is further divided into ‘Variables’ and ‘Weighting’. We hope that this improves clarity for readers.

It is better to present a diagram about selecting the provinces and hospitals.

We created a map of Thailand to show the location of the provinces where the participating hospitals were located. We refer readers to the map in the methods section on line 123.

While logistic regression models are the correct ones to use, the research does not in fact test for interactions (e.g., how gender and years of experience interact in predicting vaccine intent).

This is correct. We reviewed the list of variables included in the multiple logistic regression model and did not identify any theoretical interaction terms (where the outcome of one variable would be dependent on the outcome of another variable in the model). We clarified this in the methods section on line 174, “No interaction terms were included in the logistic regression model.”

Missing response rates: It is not known how many of the healthcare workers declined to participate or whether non-responders differed substantially from responders.

We added the participation to the results on line 179, “Overall, 2,275 health care personnel were approached, and 2,180 health care personnel participated in the survey (96% participation).

Discussion

It is well organised while just avoid from using we.

We revised the text to avoid ‘we’ and ‘our’ in several place in the discussion as noted above and especially in the limitations section on lines 397-404.

And the discussion appropriately identifies incentives and barriers but could make more comparisons with foreign literature (e.g., comparing Thai rates to rates in other countries in Asia or the world).

We conducted additional literature review to provide more context of our study findings in comparison with other studies published from East and Southeast Asian countries. We identified two systematic reviews of influenza vaccination coverage among health care personnel in Asia, mainly from East Asia. We also identified studies from other countries in Southeast Asia specifically, including Singapore and Malaysia. We identified one review on health care personnel perspectives on COVID-19 vaccination in Asia. We added a summary of these publication to the discussion section, starting on line 325.

While the study does acknowledge certain limitations, it does not take into account social desirability bias, wherein healthcare professionals may have overreported their intention to get vaccinated in an attempt to fit in with public health needs.

We revised the limitation section to acknowledge the residual effects of social desirability bias. We added on line 403, “residual social desirability bias could remain and would overestimate the willingness to receive or recommend influenza and COVID-19 vaccinations.”

It also does not take into account the impact of vaccine availability on attitudes. If COVID-19 vaccines were more accessible, it is unclear if intent to vaccinate would increase.

This is a helpful observation, which we did not explore in the current survey. We have acknowledged this possibility and the need for additional study in the discussion on lines 322-324, “Increased willingness to receive influenza vaccination compared with COVID-19 vaccination could also be related to vaccine availability, as COVID-19 vaccines are not as widely available as influenza vaccines, which could be explored further in future studies.”

Reviewer #2: General comment

This study assessed factors associated with influenza vaccination in the most recent season and intention to receive annual COVID-19 vaccination among health care personnel (HCP) in Thailand, using data from interviews conducted across 16 hospitals in eight provinces.

Given that both vaccinations are recommended for HCP in Thailand, the findings are relevant and timely, particularly in the context of efforts to improve vaccine uptake.

The manuscript is clearly written, appropriately structured, and employs suitable methodology.

While the manuscript is generally well-prepared, I would like to offer a few comments and questions that may help further improve the clarity and impact of the work.

Thank you for your careful review.

Major comments

1. I understand that the survey was conducted through interviews, but some aspects of the survey methodology remain unclear. For example, it is not clear whether the responses were selected from predefined choices or provided as open-ended answers. If predefined options were used, it seems that data could have been collected through self-administered questionnaires or an online survey instead.

It would be helpful if the authors could clarify why interviews were chosen as the method of data collection.

The survey included both multiple choice questions and questions with open-ended responses. The use of trained interviewers helped to standardize interpretation of the questions. For this analysis, however, we only analyzed the multiple-choice questions. We clarified this on lines 132-134, “Trained interviewers administered the survey questions to improve data completeness and standardize interpretation of the survey questions. Questions included multiple choice and open-ended responses; however, only multiple-choice responses were included in this analysis.”

2. The manuscript provides the rationale for the sample size calculation and the actual number of participants included; however, it would be important to also report how many individuals were initially invited to participate and how many agreed to be interviewed (i.e., the response rate).

We added the participation to the results on line 179, “Overall, 2,275 health care personnel were approached, and 2,180 health care personnel participated in the survey (96% participation).

3. The authors state that the data were weighted according to the distribution of health care personnel by occupation and hospital. However, it is not clear whether other factors—such as age, sex, or the type of patients generally treated—were also adjusted for.

Would it be necessary to consider adjusting for these variables as well? These variables were included in the multivariable model if they were associated with the outcome in the univariable analysis.

We included an explanation of variable selection in the methods section on line 173, “Factors that were significant in the univariable analysis were included in the multivariable model.”

Additionally, rather than adjusting for occupation, it may be worth considering subgroup analyses by occupational category to explore whether associations differ across groups, which could inform tailored strategies for different types of HCP. What are the authors’ thoughts on this approach?

We considered running additional analyses stratified by occupation. This would indeed be an interesting analysis with implications for implementing programmatic strategies for increasing vaccination. After careful consideration, we decided not to report subgroup analyses because our sample size calculations were not powered for individual occupational subgroups.

4. The authors state that responses were summed into scores for each perception construct using the response options ranging from 0 to 4, where 0 indicates "refuse to answer." If the numerical values were used as-is, the inclusion of 0 for "refuse to answer" may affect the validity of the score calculation.

Could the authors clarify how the scores were constructed, and specifically how "refuse to answer" were handled in the scoring process?

This is correct. Participants who responded ‘refuse to answer’ to one of the perception questions had 0 added to their score for that question. If the participant refused to answer the question, we inferred that this corresponds with low perceived disease severity, vaccine safety, or vaccine effectiveness, so we concluded that contributing ‘0’ to the score was appropriate. Fortunately, the proportion of participants who refused to answer was small (<10%) for most perception questions.

5. Table 3 includes “It was offered for free at my job (not mandatory)” and “It was free,” but the difference between them was not clear to me. It may be helpful to provide a clearer explanation in the manuscript to avoid potential confusion for readers.

We revised the table and corrected the text to clarify that “It was free” refers to free vaccine that was offered outside the hospital. The corrected text reads, “having it free at work was less frequently a motivating factor (27.8%, 95%CI 25.3–30.4).” (lines 278-279) and Table 3 was revised to “It was free (outside my workplace).”

Minor comments

1. The term "vaccine effectiveness" is used throughout the manuscript, but "vaccine efficacy" appears on page 8, line 152. Are the authors intentionally distinguishing between the two, or should the terminology be made consistent?

Thank you for catching this. We have changed “efficacy” to “effectiveness” on line 157.

2. I did not fully understand what you meant by “Proportions in the descriptive analysis that were below the National Center for Health Statistics reliability criteria were not shown [24]." on page 8, lines 164-165. Could you kindly explain?

We revised the description to clarify the reliability criteria. The reliability criteria used by the National Center for Health Statistics are complex and are explained in full detail in the accompanying citation by Parker et al. In brief, we clarified on line 169, “Proportions in the descriptive analysis that were below pre-defined reliability criteria (e.g., based on sample size, confidence interval, or degrees of freedom) were not shown.”

3. On page 18, line 300, it appears that the word "vaccination" may be missing after "COVID-19." Could the authors confirm and revise if necessary?

Correct, we revised the text to read, “If influenza and COVID-19 vaccination were offered together…” on line 305.

---

## [Decision Letter · Decision Letter 1]

22 Jun 2025

PONE-D-24-58516R1Perceptions of influenza and SARS-CoV-2 vaccination among health care personnel in Thailand, 2024

PLOS ONE

Dear Dr. Prabda Praphasiri,

Thank you for submitting your manuscript to PLOS ONE. After careful consideration, we feel that it has merit but does not fully meet PLOS ONE’s publication criteria as it currently stands. Therefore, we invite you to submit a revised version of the manuscript that addresses the points raised during the review process.

We look forward to receiving your revised manuscript.

Kind regards,

Sirwan Khalid Ahmed

Academic Editor

PLOS ONE

Journal Requirements:

Reviewers' comments:

Reviewer's Responses to Questions

**Comments to the Author**

1. If the authors have adequately addressed your comments raised in a previous round of review and you feel that this manuscript is now acceptable for publication, you may indicate that here to bypass the “Comments to the Author” section, enter your conflict of interest statement in the “Confidential to Editor” section, and submit your "Accept" recommendation.

Reviewer #1: All comments have been addressed

Reviewer #2: (No Response)

2. Is the manuscript technically sound, and do the data support the conclusions?

Reviewer #1: Yes

Reviewer #2: Yes

3. Has the statistical analysis been performed appropriately and rigorously? 

Reviewer #1: Yes

Reviewer #2: Yes

4. Have the authors made all data underlying the findings in their manuscript fully available?

Reviewer #1: Yes

Reviewer #2: Yes

5. Is the manuscript presented in an intelligible fashion and written in standard English?

Reviewer #1: Yes

Reviewer #2: Yes

6. Review Comments to the Author

Reviewer #1: Many thanks for following my recommendations. Right now the paper it looks good and it is ready to publication.

Reviewer #2: General Comments

The authors have generally addressed my previous comments appropriately.

However, I would like to offer a couple of follow-up remarks regarding their responses to my earlier comments.

These are detailed below.

Comment 1

Thank you for your response and the revisions made to the manuscript.

Based on your reply, I understand that the open-ended responses were not included in the analysis.

Could you clarify what kind of questions generated these open-ended responses?

Additionally, what was the rationale for excluding them from the analysis?

Even if you judged that analyzing them was not necessary for the purpose of this study, it would be helpful to provide a general overview of the survey content — specifically, the exact questions that were asked in both the multiple-choice and open-ended formats, and which of these were included in the analysis. This information can help readers better understand how the respondents perceived the survey and the potential cognitive or time burden involved in completing it.

I also suggest that you consider briefly stating the reason for excluding the open-ended responses in the revised manuscript. Furthermore, providing a full list of the survey questions — including those not analyzed — as a Supplementary File may help convey the overall scope and structure of the survey more transparently.

Comment 2

Thank you for your explanation regarding how you handled the “refuse to answer” responses.

I agree that such responses might reasonably correspond to low perceived disease severity, vaccine safety, or vaccine effectiveness, as you suggest. However, this assumption may not always hold true in every case. In some cases, it is also possible that the respondent had a high level of perception and chose not to answer intentionally for personal or strongly held reasons.

I also understand that the proportion of such responses was small (<10%) for most perception questions. Nevertheless, given that the perception score serves as a key component in summarizing the survey results in your study, I believe it would be helpful to provide this information for readers’ understanding.

I suggest that you clearly state in the manuscript the rationale for assigning a score of 0 to “refuse to answer” responses. If you agree, it would also be worth briefly acknowledging that this approach may not fully capture the range of possible interpretations for such responses. Finally, I recommend including a note in the manuscript indicating that the proportion of “refuse to answer” responses was low.

7. PLOS authors have the option to publish the peer review history of their article (what does this mean? ). If published, this will include your full peer review and any attached files.

**Do you want your identity to be public for this peer review?** For information about this choice, including consent withdrawal, please see our Privacy Policy .

Reviewer #1: **Yes: ** Assist. Prof. Dr. Kochr Ali Mahmood

Reviewer #2: No

---

## [Author Response · Author response to Decision Letter 2]

14 Jul 2025

Thank you for the opportunity to make additional revisions to improve our manuscript. We have addressed the reviewer’s two comments and provided a point-by-point response to each comment below.

Reviewer #2: General Comments

The authors have generally addressed my previous comments appropriately.

However, I would like to offer a couple of follow-up remarks regarding their responses to my earlier comments.

These are detailed below.

Comment 1

Thank you for your response and the revisions made to the manuscript.

Based on your reply, I understand that the open-ended responses were not included in the analysis.

Could you clarify what kind of questions generated these open-ended responses?

Additionally, what was the rationale for excluding them from the analysis?

Even if you judged that analyzing them was not necessary for the purpose of this study, it would be helpful to provide a general overview of the survey content — specifically, the exact questions that were asked in both the multiple-choice and open-ended formats, and which of these were included in the analysis. This information can help readers better understand how the respondents perceived the survey and the potential cognitive or time burden involved in completing it.

I also suggest that you consider briefly stating the reason for excluding the open-ended responses in the revised manuscript. Furthermore, providing a full list of the survey questions — including those not analyzed — as a Supplementary File may help convey the overall scope and structure of the survey more transparently.

Response 1

The open-ended questions provided survey participants the opportunity to provide additional explanation to the multiple-choice questions. We have added a copy of the survey in Thai and English as a supplemental file for readers to review. We have revised the text to provide additional explanation on our decision to not analyze the open-ended response questions in this analysis. On line 134, we explain, “Open-ended responses were used to provide additional clarification to multiple choice responses and were not systematically analyzed.” We also included a reference to the supplemental file on line 104.

Comment 2

Thank you for your explanation regarding how you handled the “refuse to answer” responses.

I agree that such responses might reasonably correspond to low perceived disease severity, vaccine safety, or vaccine effectiveness, as you suggest. However, this assumption may not always hold true in every case. In some cases, it is also possible that the respondent had a high level of perception and chose not to answer intentionally for personal or strongly held reasons.

I also understand that the proportion of such responses was small (<10%) for most perception questions. Nevertheless, given that the perception score serves as a key component in summarizing the survey results in your study, I believe it would be helpful to provide this information for readers’ understanding.

I suggest that you clearly state in the manuscript the rationale for assigning a score of 0 to “refuse to answer” responses. If you agree, it would also be worth briefly acknowledging that this approach may not fully capture the range of possible interpretations for such responses. Finally, I recommend including a note in the manuscript indicating that the proportion of “refuse to answer” responses was low.

Response 2

The response of refusing to answer is indeed ambiguous. We have revised the manuscript as suggested in the limitations paragraph on lines 402-406. We provide some explanation of our rationale and acknowledge the limitations of this approach. This reads, “For perception questions, participants who refused to answer were assigned a score of 0, as we assumed this response corresponded with low perceived disease severity, vaccine safety, or vaccine effectiveness; however, this response is ambiguous, and a score of 0 might not be appropriate in all cases. Overall, the proportion of participants who refused to answer was small (<10%) for most perception questions.”

---

## [Editor Report · Decision Letter 2]

17 Jul 2025

Perceptions of influenza and SARS-CoV-2 vaccination among health care personnel in Thailand, 2024

PONE-D-24-58516R2

Dear Dr. Praphasiri,

We’re pleased to inform you that your manuscript has been judged scientifically suitable for publication and will be formally accepted for publication once it meets all outstanding technical requirements.

Kind regards,

Sirwan Khalid Ahmed

Academic Editor

PLOS ONE
---

## [Editor Report · Acceptance letter]

PONE-D-24-58516R2

PLOS ONE

Dear Dr. Praphasiri,

I'm pleased to inform you that your manuscript has been deemed suitable for publication in PLOS ONE. Congratulations! Your manuscript is now being handed over to our production team.

Kind regards,

on behalf of

Dr. Sirwan Khalid Ahmed

Academic Editor

PLOS ONE